# Protocol for Designing New Functional Food with the Addition of Food Industry By-Products, Using Design Thinking Techniques—A Case Study of a Snack with Antioxidant Properties for Physically Active People

**DOI:** 10.3390/foods10040694

**Published:** 2021-03-24

**Authors:** Joanna Tkaczewska, Piotr Kulawik, Małgorzata Morawska-Tota, Marzena Zając, Paulina Guzik, Łukasz Tota, Paulina Pająk, Robert Duliński, Adam Florkiewicz, Władysław Migdał

**Affiliations:** 1Department of Animal Product Technology, Faculty of Food Technology, University of Agriculture in Kraków, al. Balicka 122, 30-149 Kraków, Poland; kulawik.piotr@gmail.com (P.K.); marzena.helena.zajac@gmail.com (M.Z.); paulinag93@o2.pl (P.G.); wladyslaw.migdal@urk.edu.pl (W.M.); 2Department of Sports Medicine & Human Nutrition, University School of Physical Education in Kraków, al. Jana Pawla II 78, 31-537 Kraków, Poland; mm.morawska@wp.pl; 3Department of Physiology and Biochemistry, Faculty of Physical Education and Sport, University of Physical Education in Krakow, al. Jana Pawla II 78, 31-537 Kraków, Poland; lukasztota@gmail.com; 4Department of Food Analysis and Quality Assessment, Faculty of Food Technology, University of Agriculture in Kraków, ul. Balicka 122, 30-149 Kraków, Poland; paulina.pajak@urk.edu.pl (P.P.); adam.florkiewicz@urk.edu.pl (A.F.); 5Department of Biotechnology and General Technology of Food, Faculty of Food Technology, University of Agriculture in Kraków, ul. Balicka 122, 30-149 Kraków, Poland; robert.dulinski@urk.edu.pl

**Keywords:** design thinking, food design, protocol, functional food, food by-products, protein hydrolysate

## Abstract

The aim of the work was to develop an easy-to-follow protocol for designing novel functional products with the addition of food industry by-products using design thinking techniques. As a result, a 12-step protocol has been designed and presented. The protocol consists of steps from the initial formation of the design team, through all the stages of the production and prototyping, until establishing the final storage conditions and creating final documentation. The protocol has been validated and explained using a case study in which a fish industry by-product hydrolysate with bioactive properties was used to develop a novel functional food product for physically active people: a date bar with carp meat and carp skin gelatin hydrolysate. Following the 12 steps presented in the protocol resulted in developing a food product with high nutritional value and antioxidant power which remains stable during storage at reduced temperatures. Moreover, the product is characterized by good sensory qualities and can be easily implemented into full-scale production. The newly designed protocol is an easy-to-follow method that could be used in almost any kind of food industry sector to sucesfully develop user-focused functional food products with by-product addition.

## 1. Introduction

The design of novel food products can be a challenging and often risky enterprise. Design thinking, a method that focuses problem solving on humans rather than on technology or organization, is becoming a very important approach among the designers in many fields [1,2], including food design [3]. The aim of the method is to create products or services that are user-oriented but, at the same time, economically viable and technologically feasible. The typical design thinking protocol consists of (i) observation and synthesis, (ii) visualisation and rapid prototyping, and (iii) revising and refining [4]. According to Olsen [3], despite the growing interest in the use of design thinking for designing novel food products, there is a lack of proper scientific articles related to this topic and food. The author has published three aspects that should be considered when designing new food products using the design thinking method. The first aspect is to use consumer empathy, which basically involves knowing your customer/user. This includes immersing the designers into the lives of their target consumers and performing early consumer testing of the designed product. The second aspect is visualization and rapid prototyping, which involves creating models or sketches and, if possible, the early version prototypes. This allows the gathering of quick feedback on the strengths and weaknesses of the product and enables the implementation of the necessary corrections before significant investments in the product are made. The third aspect is described as collaboration, which basically means involving many different actors into the design procedure. This means a collaboration between, for example, the scientists and the industry, but also that there is a need to create a team of designers with different backgrounds and specializations. Design thinking uses and implements many various tools that can be used in the designing process. A few examples include multidisciplinary teams, feedback, brainstorming, storytelling, multiple design teams, time constraints, iteration, prototyping, watching out for extreme users, etc. [5].

The food industry generates an enormous amount of waste, estimated at around 1.6 billion tons and approximately 1 trillion USD losses yearly [6]. Many of those food wastes can be reused and revalorized obtaining biomass, biofuels, biofertilizers, or biologically active compounds [7]. One of the most important revalorized food wastes are protein hydrolysates [8]. Protein hydrolysates are obtained through enzymatic hydrolysis, which entails the use of enzymes to break down proteins and alter the chemical, functional, and sensory properties of proteins without decreasing their nutritional value. Protein hydrolysates hold great promise as valuable functional ingredients in healthy diets [9].

Food industry by-products are becoming of great interest as ingredients of novel functional food products [10]. Functional food is derived from naturally occurring substances, which can and should be consumed as part of one’s daily diet and further serves to regulate or otherwise affect a particular body process when ingested [11]. As opposed to many novel food products being designed, according to Roberfroid [12], the development of new functional products should remain, first and foremost, a scientific and not a marketing challenge. This is the condition for success in both human health and the food industry.

Currently, attention is being paid to the role of physical activity in maintaining health. There is also greater interest in dietary supplements. The increase in their consumption is not only limited to athletes, but to individuals recreationally undertaking physical activity [13]. A significant group of supplements are preparations designed to reduce oxidative stress in the body of athletes. This is important because oxidative stress is a consequence of a disturbed pro-oxidant–antioxidant balance in conditions of vigorous physical exercise, mainly developing as a result of intensified metabolic processes, but also due to environmental factors. Oxidative stress contributes to an increased demand for antioxidant vitamins and bioflavonoids among athletes [14].

Due to the potential anti-inflammatory and antioxidant properties of fish, their regular consumption by athletes is encouraged [15]. Cyprinids (*Cyprinidae*) are major sources of animal protein for millions of people in many Asian countries and are an appreciated food in eastern Europe [16]. Carp (*Cyprinus carpio* L.) production in Europe in 2016 was 4,328,083 tonnes and is experiencing a downward trend Adámek [16]. The main reason for the decrease in market demand for carp is that it is traditionally offered in the form of whole fish, which is losing against competition with highly processed and more convenient products from other fish species. To increase the range of carp products, producers are looking for alternatives to fresh carp; thus, processing of this fish is beginning to dynamically develop [17].

Although, as mentioned previously, Olsen [3] has pointed out three important aspects in designing novel food products, to the authors’ knowledge, there is no existing protocol in the scientific literature that would provide detailed guidelines for designing novel functional food products by implementing the design thinking approach. Moreover, since this approach is very general in its description, it may often be difficult to use by non-experienced designers. Therefore, the aim of this study was to develop a detailed step-by-step protocol for designing functional food products with the addition of food industry by-products, using the design thinking approach to create new functional food products. To visualize the use of the protocol, all the steps are described using the example of a case study—a novel pre-exercise snack food with antioxidant properties containing carp skin gelatin hydrolysate (CSGH).

## 2. Materials and Methods

### 2.1. Protocol Development

The protocol was developed taking into consideration the traditional problem solving operation with a combination of the design thinking methodology, including three key phases: inspiration, ideation, and implementation, as described by Serrat [18].

### 2.2. Materials Used during Case Study

Carp skins (*Cyprinus carpio*), a waste product from industrial filleting, were obtained from Sona Sp. z o.o. (Koziegłowy, Poland). The skins were ground after the removal of muscle tissue and scales (Mado MEW 613, Germany). The extraction of gelatine from the minced skins was performed as described by Tkaczewska et al. [19]. The hydrolysate with the highest antioxidant properties was obtained by following the instructions described by Tkaczewska et al. [20]. CSGH was freeze-dried, ground, and stored as a powder without access to humidity or light.

Carp fillets were obtained from the Sona Sp. z o.o. fish processing plant (Koziegłowy, Poland). Nuts, seeds, and dates were purchased from a local supplier (Perlo, Gorlice). The equipment used during the prototype snack production included: the Mado MEW 613 grinder (Dornhan, Germany), R2 Robot Coupe cutter (Vincennes, France), and Mainca RM-20 mixing machine (Barcelona, Spain).

### 2.3. Analysis of Nutritional Value

The dry weight, lipid, ash, and protein content were determined via methods recommended by the Association of Official Analytical Chemists [21]. The cholesterol content was determined using the colorimetric method by Röschlau et al. [22]. The analysis was performed using the R-Biopharm kit (Cat. No. 10139050035, Darmstadt, Germany).

Amino acid analysis was performed using the Dionex Ultimate 3000 HPLC system (Thermo Fisher Scientific, Waltham, MA, USA) equipped with an LPG-3400 SD four-channel gradient pump, WPS 3000 TSL auto-sampler, and FLD 3400RS four-channel fluorescent detector via the method described by Tkaczewska, Borawska-Dziadkiewicz, Kulawik, Duda, Morawska, and Mickowska [20].

The detection of B1 and B2 vitamins (mg/kg of dry matter) was performed as described in Starzyńska-Janiszewska et al. [23], with modifications for thiamine detection by Mickowska et al. [24]. Vitamin B12 was determined using the Biopharm test (Cat. No. R2103, Germany) on an Elisa Sunrise™ microplate reader (Tecan, Männedorf, Switzerland).

Selected mineral compounds—calcium, magnesium, potassium, and sodium—were analysed using the FAAS flame atomic absorption spectrometric method (Varian AA240FS) according to PN-EN 15505:2009, and zinc and manganese according to PN-EN 14084:2004. All the applied methods were fully validated and checked by internal quality control procedures according to PN-EN 13804 and inter-laboratory/proficiency tests.

### 2.4. Sensory Evaluation

A sensory evaluation of the snacks was performed at the Laboratory for Sensory Analysis of Food, Faculty of Technology, University of Agriculture in Kraków, meeting PN-EN ISO 8589:2010 requirements. The sensory analysis used during the case study involved different protocols at different design steps.

The oranoleptic assessment performed by the design team members included five panellists. The scores were individually assigned by each panellist and discussed after every member finished their evaluation. The purpose of organoleptic evaluation was to reject products that were obviously not of appropriate sensory quality and did not meet the requirements defined by the members of the project team. The members of the project team organised a brainstorming session, during which the possible preferences of consumers from the target group were discussed, as well as sensory features that may be subjectively important when evaluating this type of product. On this basis of and with regard to data from the literature [25], qualitative characteristics were selected that are significant for snack products, such as: appearance, colour, texture, flavour, taste, and overall acceptability. All the differentiators were assessed on a simple five-point hedonic scale: 1—very bad, 2—bad, 3—neither good nor bad, 4—good, 5—very good. The products with the lowest ratings were rejected by the design team.

The consumer sensory analysis included a specific number of panellists depending on the stage of design recommended by Moskowitz [25] for the early-stage design testing of food products. Consumer analyses were conducted twice. The first consumer analysis of the selected snack prototype was carried out among 20 consumers. The research group consisted of participants (30% male and 70% female) aged 22 to 55, living in large cities (100,000 inhabitants or more), who eat fruit bars at least once a month and are physically active from time to time.

A consumer analysis of the snack stored in refrigerated conditions for six weeks was carried out with the participation of 50 panelists. The consumers were mostly male (76% men and 24% women, age 21 to 35 years) studying at the Univeristy of Physical Education, demonstrating high levels of physical activity (undertaking physical activity at least five times a week). A total of 30% of the respondents lived in large cities (over 100,000 inhabitants), 42% in small towns (20,000 to 100,000 inhabitants), and 28% in villages. The recruited panelists were unrelated to the product design process and unaware of the product production process as well as the ingredient list. The consumer assessment was carried out using the method of a nine-point hedonic scale: 1—greatly disliked, 2—disliked very much, 3—moderately disliked, 4—disliked a bit, 5—neither liked nor disliked, 6—moderately liked, 7—liked a bit, 8—liked very much, 9—greatly liked.

The sensory evaluation by the professional sensory panel included a group of 12 panelists trained according to PN-ISO 3972:2016-07, PN-EN ISO 11132:2017-08, and PN-EN ISO 8586:2014-03. The panelists were given detailed instructions concerning the way of assessing the snack attributes. For the five-point evaluation, cards characterising flavour (general appearance, colour, smell, texture in the mouth, taste, overall taste impression) were created. For assessment using the profiling method, a list of qualitative features established during a special session was used: smell, colour, texture, and taste. The list included 24 attributes (eight smells, three colours, five textures, and eight flavours).

### 2.5. Analysis during Shelf-Life Evaluation

During the evaluation of the shelf-life and storage conditions, the snacks were packed in PET trays with covers and stored under the following conditions: (i) room temperature of 20 °C for two weeks, (ii) chilled at +4 °C for six weeks, and (iii) frozen at −18 °C for six months.

Analysis regarding the quality and antioxidant properties of the snacks was performed every seven days (room temperature and chilled storage) or every 14 days for the first month, and then every four weeks up to six months (frozen storage).

#### 2.5.1. Antioxidant Activity

Chemical extraction of the antioxidants from the innovative snack with CSGH was executed according to the method described by Pérez-Jiménez and Saura-Calixto [26].

Determination of sample reducing potential was performed according to the method described by [27], including modifications. The oxidant in the FRAP assay consisted of an acetate buffer (pH 3.6), ferric chloride solution (20 mM), and 2,4,6-tripyridyl-s-triazine solution (10 mM TPTZ in 40 mM HCl) at 10:1:1 (*v*/*v*/*v*), respectively, which was freshly prepared on the day of analysis. The FRAP assay was performed by incubating the sample and reagent at 37 °C for 4 min. Absorbance at 593 nm was determined relative to a reagent blank also incubated at 37 °C using the Helios Gamma UV-1601 spectrophotometer (Thermo Fisher Scientific, Waltham, MA, USA). The results were calculated as μmol of Trolox equivalent per 1 mg of snack wet weight.

The ability to scavenge the 2,2-diphenyl-1- picrylhydrazyl (DPPH) free radical was measured using of the method by Ryan et al. [28].

After incubation, absorbance was measured at 517 nm. Scavenging effect was calculated using the formula:
(1)$$\mathrm{DPPH}\;\mathrm{radical}\;\mathrm{scavenging}\;\left(\%\right)=\;\frac{A_{\mathrm{blank}}-{\;A}_{\mathrm{sample}}}{A_{\mathrm{blank}}}\times 100\%$$

#### 2.5.2. Microbiological Quality

Ten grams of the ground sample was aseptically weighed into a stomacher bag, diluted using 90 mL of sterile buffered peptone water (Biomaxima, Warsaw, Poland), and homogenised using a Stomacher device for 3 min. The homogenate was used to prepare further dilutions.

Total viable count (TVC) analysis was conducted using the plate count agar (Biomaxima, Poland) via the pour plate method. Incubation took place for 48 h at a temperature of 30 °C. Yeast and mould were analysed with the spread plate method using DRBC agar (Biomaxima, Poland) and incubated for five days at 25 °C. All the results were presented as log10 of colony-forming units per gram of the sample.

#### 2.5.3. TBARS Analysis

For Thiobarbituric acid reactive substances (TBARS) assay, 10 g of a homogenised sample was mixed with 34.25 mL of 4% perchloric acid and 0.75 mL of 0.01M BHT in ethanol. The solution was homogenised and centrifuged for 15 min at 10.000 × *g*. Afterwards, the solution was filtered through Whatman No. 1 filtering paper into a volumetric flask and filled with 4% perchloric acid up to 50 mL. Then, 5 mL of the filtrate was transferred into a tube with 5 mL of 0.02 M TBA solution. If needed, the filtrate was additionally diluted with 4% perchloric acid before being mixed with TBA to obtain absorbances of the final sample between 0.1 and 0.7. As the filtrate was not colourless, control samples were prepared containing filtrate (diluted with 4% perchloric acid if needed) and distilled water. Later, the samples were incubated in a water bath at 90 °C for 60 min in the dark, cooled down under running water and measured for absorbance at 532 nm using the Helios Gamma spectrophotometer (Thermo Fisher Scientific, Waltham, MA, USA). TBARS was calculated from the formula:
(2)$$TBARS\;\left[\mathrm{mg}\;\mathrm{MDA}/\mathrm{kg}\right]=\frac{\left(A-C\right)\cdot K}{B\;\cdot \;0.2}$$
where *A* is the absorbance of the sample, *C* is the absorbance of the control, *K* is the extension coefficient (5.5), *B* is the filtrate amount (mL), and 0.2 is the sample amount in mL of the filtrate (mg).

#### 2.5.4. Texture Analysis and Water Activity

Instrumental texture analysis was performed using the TAX-T2 Plus texturometer (Stable Micro Systems, UK). All the bars were tested at room temperature (20 °C). The bars—7 cm × 2 cm × 2 cm—were penetrated to 50% of initial height using a P/6 cylindrical probe at a test speed of 2.0 mm/s. Each bar was punctured eight times.

The water activity of the snack was measured using the Novasina LabMaster water activity metre (Novasina, Switzerland) under controlled temperature.

### 2.6. Statistical Analysis

All analyses were performed using triplicate independent repetitions. Analysis on each replicate was carried out in duplicate unless otherwise stated. The statistical analysis was performed using Statistica 13.1 software (Tibco, Palo Alto, CA, USA). The normality of the results was tested using the Shapiro–Wilk test and the Box–Cox transformation was performed on variables with non-normal distribution. The differences in the results of snacks during storage in different conditions was analysed using two-way ANOVA, where the storage temperature and day of storage were used as independent variables. Tukey’s test was used to establish differences between individual groups and a 95% confidence interval was chosen (*p* < 0.05).

## 3. Results and Discussion

The created protocol for developing functional food products with the addition of food industry by-products is presented in (Figure 1).

The protocol starts with the formation of the design team and covers 12 steps of conduct which should allow the creation of the final food product available for production and implementation. Although the protocol is presented in a linear manner, for ease of reading, it is important to remember that the product development is actually a non-linear process, and the design team should sometimes take a few steps back to revise the previously made assumptions or even start the whole process from the beginning.

When forming a design team, special care should be taken to choose members with different areas of specialisation and experience. This ensures diverse opinions and means to solve problems that arise during the design and prototyping. The team can accept additional members during the design process if such a need arises.

Case study: The research facility, after consulting with various fish processing facilities located in Poland, was given the task of developing a novel functional food product containing carp (*Cyprinus carpio)* meat and CSGH, a novel bioactive ingredient developed from carp processing by-products [29]. The product had to be innovative in its form and was supposed to be part of the solution to the problem of the drastic decrease in carp meat consumption among Polish consumers. The research facility formed a design team of three food technologists with the sole purpose of designing a novel functional food product. The first member specialised in functional food ingredients, such as protein hydrolysates and bioactive peptides, with working experience in dietetics. The second specialised in food security and safety, while the third was a specialist in the area of nutritional value and had working knowledge and experience in the industrial-scale production of food products.

### 3.1. Step 1. Identification of the Target Consumer Group

Although many design teams start their product development by defining the product and then establish to whom the product could be directed to, design thinking, due to its focus on the user, requires a completely different approach. The first step of the design protocol is to determine the target population group, towards which the newly developed product is being directed. This allows for tailoring the food product directly to the needs of consumers and not the other way around, trying to fit consumer needs into the previously designed product. Narrowing the size of the target group allows for easier diagnosis of the problems for such a group during the following steps. The target consumer group is often provided by marketing specialists of a company that develops new products or can be obtained through market research.

Case study: The design team, based on the current market situation, decided that the target consumer group should be physically active people—both amateurs and professionals.

This is a rapidly developing consumer group that is willing to purchase novel food products and is actively seeking out functional foods [30].

After establishing the target consumer group, the design team decided they lacked the appropriate knowledge about sports nutrition and decided to recruit additional members to the design team—sports nutrition specialists.

### 3.2. Problem Diagnosis

During the next step, the design team should focus on problems faced by the target group. The establishment of problems should be user-oriented and apply empathy. A detailed guide about the use of an empathy-based approach during product design has been provided by Dalton and Kahute [31]. In brief, one of the most important aspects that should be considered is that one cannot talk to himself, meaning that, aside from trying to empathise with a hypothetical target group member to list his/her problems, the design team members should have direct contact and conversations with actual target group members. This often allows designers to be more aware of the potential problems and the needs of the target group and to discard incorrect assumptions that could have been made.

Case study: The design team, having created a list of already known or assumed information about the physically active people, start to collect the information in the field: interviewing professional athletes that the members had contact with, going to fitness clubs and gyms where they observed and conversed with other club members, talking to their friends and family, and going to city parks frequently visited by people undertaking physical activity during their leisure time. Afterwards, the design team performed a comparison of their notes and insights and brainstormed about various problems that they had identified. After consideration and cross-checking, a few problems were selected: (i) time—most of the physically active people had limited time during their day/week to perform physical activity; (ii) fatigue—some people reported fatigue, mainly due to daily life activities but also due to over-exercising with little amounts of time between exercises for regeneration; (iii) low training efficiency—some respondents complained that their training is not as efficient as they would like it to be and also that the improvement in stamina and endurance due to regular workouts is below their expectations; (iv) high expenses—respondents mentioned that their fitness requires substantial financial outlays, which are spent on memberships at fitness clubs and various dietary supplements.

### 3.3. Identification of the Source of the Problem

After identifying the main problems, it is necessary to go to their sources. This can be achieved by various root-cause analysis techniques, which have been extensively described in the existing literature. These techniques include such methods as the “five whys”, fishbone diagrams, Pareto diagram, decision tables, casual trees, etc. [32,33]. When the source of the problem has been identified, the design team has to determine if it can be, at least partially, solved by a food product. If not, the whole process of problem diagnosis has to be repeated. On the other hand, it might also occur that multiple problems can be solved by various food products. In such a case, the design team may take all of this information into consideration during the next steps of the protocol.

Case study: The design team used the “five whys” technique for root-cause analysis of the identified problems; however, due to the increased number of branches appearing during each consequent “why”, the fishbone diagram was also used. As an example of typical “five whys” analysis, the examination of a low-training efficiency problem is presented: People experience low training efficiency. Why? (1): Inappropriate post-exercise regeneration. Why? (2): Too high post-exercise oxidative stress (different responses possible with branching, e.g., inappropriate nutrition or too short a rest between exercises, etc.—an additional five whys should be performed for each branching). Why? (3): Insufficient efficiency of the antioxidant system. Why? (4): Insufficient levels of antioxidants. Why? (5): Inappropriate diet. The analysis was based on both the design team members’ experience and the scientific literature [34].

After analysis, the design team concluded that each of the four identified problems can be at least partially solved by implementing appropriate food products.

### 3.4. Creating the Food Product Concept

Knowing the target group and based on the interviews performed in Step 2, and identifying the problem sources in Step 3 allows the definition of the key features of the designed product. During this step, the design team does not yet propose a specific product, but only the requirements it must fulfil.

Case study: The design team, based on the identified problems and their sources, listed key features of the product:(a)The product should have high antioxidant potential (response to problems (ii)—fatigue, and (iii)—low training efficiency);(b)The product, used as a pre-exercise snack, should be properly balanced regarding the amount of supplied energy (200 to 300 kcal/100 g), proteins, carbohydrates, and fat (response to problems (ii)—fatigue, and (iii)—low training efficiency);(c)As a pre-workout snack, the product should contain as many carbohydrates as possible. The snack protein and fat levels should not exceed 20 and 25% of the total energy, respectively (response to problems (ii)—fatigue, and (iii)—low training efficiency);(d)The protein contained in the snack should be complete with high nutritional value (response to problems (ii)—fatigue, and (iii)—low training efficiency);(e)The product should contain both simple and complex carbohydrates, for the constant replenishment of blood glucose levels (response to problem (iii)—low training efficiency);(f)The product should consist of natural ingredients, limiting food additives as much as possible, remaining attractive to consumers following the “clean label” trend (marketing advantage);(g)If possible, the snack should consider new trends among consumers, i.e., gluten-free or lactose-free products (marketing advantage and broader consumer range);(h)The product should have the simplest possible production process for easy implementation into the production line (economic feasibility advantage);(i)The product should contain the largest possible share of carp meat, while, at the same time, having high sensory quality (fundamental requirement of the requester);(j)The product should include CSGH, which is a source of biologically active peptides with antioxidant properties (fundamental requirement of the requester).

Such food products may address two of the four identified problems of the customers (fatigue and low training efficiency).

### 3.5. Proposed Food Products

After establishing the key features, the design team should create a list of food products that meet the requirements established in Step 4. A possible method would be to perform brainstorming, and afterwards, to carefully evaluate if all the key features that can be tested at the current stage are met by the proposed food product. If yes, then the food product is added to a list of products for preliminary prototyping. If not, then the product idea should be modified or discarded. It should be noted that, at this stage, products are not discarded due to the expected sensory properties—this is evaluated during the early prototyping stage.

Case study: Each design team member was given a few days to consider and propose a list of possible pre-workout snacks that would contain both CSGH and carp meat, which resulted in proposing 10 food products. After all the product propositions were presented, the design team discussed and evaluated their compliance with key features; three food snacks remained: date bars with carp meat and CSGH, cereal bars with carp meat and CSGH, and baked wheat fingers with carp meat and CSGH.

### 3.6. Preparation of Preliminary Prototypes

One of the key concepts of design thinking is to prototype at the early stages of development. The design team should prepare preliminary recipes and try to create the first prototypes of the food products, which should first be subjected to organoleptic analysis and then additional analysis if required. The first organoleptic analysis can be performed by the design team members to eliminate the products which obviously will not be of appropriate sensory quality and can be further confirmed by analysis on consumers or a professional sensory panel if required. Additional sensory testing at this stage increases the design costs, but it might speed up the development process; therefore, the best approach, depending on the circumstances, should be adopted by the design team. The additional analysis that can be performed depends on the type of functional food product that is to be developed. It can include analysis confirming functional properties such as antioxidant or antihypertensive qualities, the amount of certain selected nutrients, or others, depending on the required key features. The design team should also establish the threshold levels for each measured parameter to assess if the product fits the requirements. If none of the products chosen in Step 5 meet the requirements, the design team has to go back to Step 5 and propose new products. In such a case, it may be necessary to hire more team members, conduct additional interviews with potential users or take inspiration from the products already present on the market.

At the end of this step, the design team should choose the best fitting food product for further development. Since the purpose is to design functional food products, sensory scores should not be the only determining factor during the selection. Therefore, if few products have been selected as appropriate, the design team should choose the best one, evaluating the entirety of the obtained results.

Case study: The design team created the preliminary recipes for three selected snacks and created early-stage prototypes. Each snack was subjected to organoleptic analysis performed by the design team members, who afterwards discussed the possibility of quick improvement. After that, the upgraded early prototypes were created and tested once more. The greatest challenge was to combine carp meat with other ingredients, which usually are not used in the same product. The cereal bars with carp meat were discarded at the early sensory evaluation stage. The date bars with carp meat also received low sensory scores during the first organoleptic evaluation; however, during the discussion, the design team decided to introduce spices into the formulation and to check different methods of heat treatment for carp meat. From the different heat treatment methods and parameters tested, baking at a temperature of 170 °C until the centre of the product reached 77 °C resulted in improvement of the sensory scores. The prototype was also produced again, with the inclusion of the following spices: coriander, chilli pepper, cloves, vanilla, cardamom, and cinnamon. Out of those, the addition of cinnamon significantly helped to improve the sensory scores. Both modifications resulted in obtaining a snack with acceptable sensory qualities.

The highest organoleptic scores were obtained for baked wheat fingers with carp meat; however, this product, when analysed for antioxidant properties, showed much lower FRAP values of 2.05 ± 0.38 µM trolox/mg compared to 12.32 ± 1.5 µM trolox/mg found in date bars with carp meat. Therefore, due to comparable final sensory scores, higher antioxidant properties and the less complicated production process, the design team has chosen date bars with carp meat as a snack for further development.

### 3.7. Creating a Product Description Sheet for the Chosen Food Product

The next step was to develop a product description sheet for the designed product. This should include information about the proposed shape, weight, and size of the product, a list of ingredients and, if relevant, other physicochemical properties, such as odour, colour, texture, etc. In this step the design team should also determine the no observable adverse effect level (NOAEL) and acceptable daily intake (ADI) levels of the functional ingredient to be used. Those can already be provided by the existing law regulations or literature, or have to be established by the design team. There is a vast amount of literature regarding the protocol for establishing both critical limits [35,36]. During this step, the design team should also consider local legislative issues to ensure that no unexpected barriers occur at the time of later commercialization. Since legislation issues usually vary from country to country, the design team should, in the first place, consider the markets in which the product will be appearing [37].

Case study: The product should have the shape of a prism with a rectangular base. A spherical shape was considered; however, this would complicate the production process, reduce storage ergonomics and be more complicated for consumption (cuboid allows to pull out only a fragment of the snack from the packaging, maintaining clean hands). The weight of the product should not be too high, so as not to cause a feeling of fullness for people during exercise, or too small, because the product must provide a certain amount of carbohydrates. The weight was established within the range of 50 to 75 g, with the exact weight established after calculating the nutritional value of the snack in the following steps. The list of ingredients included dried dates, baked carp meat, oat flakes, walnuts, sunflower seeds, cinnamon, and CSGH.

The snack had to contain CSGH, which does not yet have NOAEL or ADI established. Both limits were determined based on research among laboratory animals [38,39]. Studies on young Wistar rats consuming CSGH in the quantity of 1 and 10% of the total diet over a six-week period showed that NOAEL of the hydrolysate was 1% for the rat diet, which, after considering the weight of the rats, amounted for 0.85 g/kg body mass. The ADI dose for humans was calculated including a 10-fold safety margin due to species-related differences and an additional 10-fold safety margin due to individual hypersensitivity within a given species. The ADI for humans, after adopting a 100-fold dose reduction, was 8.5 mg CSGH/kg of body mass. Analysis of local legislation showed that the product will have to be classified as a so-called novel food as defined in EU Regulation 2015/2283, due to the fact that CSGH is a food constituent that has been developed in recent years and there is no history of its consumption before 1997. Therefore, at the end of the development process, the company will have to apply to the European Commission for authorisation of the novel food. However, this would require subjecting the CSGH to additional safety studies, such as allergenicity, genotoxicity, chronic and subchronic toxicity, carcinogenicity, and reproductive and developmental toxicity as required in the EFSA guidelines [37,40]. Subjecting the CSGH to the novel food database also requires providing information regarding details of the processing, including production flow charts and quality and safety assurances, such as GMP and/or HACCP. However, since the design team was tasked with developing a new food product which included CSGH, and not with establishing the additional safety properties of CSGH alone, the required safety studies will have to be performed by the food industry entity that will be producing CSGH.

### 3.8. Creating a Detailed Recipe and the Product Production Process

The next step involves creating a detailed recipe of the product along with the production process. The nutritional value of the product may also be calculated during this step, which might result in modification of the recipe and/or technological process. After the ADI of the functional ingredient has been established, the appropriate amount of functional ingredient should be included into the list of ingredients, taking the planned weight of the food product and the level of possible daily consumption into consideration.

Case study: Based on a food product nutritional value database [41], the nutritional value of the snack was estimated and the recipe was modified so that the product would meet the established key features. Two modifications were implemented: (i) walnuts were replaced with pecan nuts due to their higher oxidative potential [42]; (ii) oat flakes were replaced with buckwheat flakes because they are gluten-free and have greater antioxidant potential [43]. The appropriate product weight was established as 70 g. This allows the determination of a dose of CSGH in the product at the level of up to 1%, without exceeding the ADI for a human weighing 70 kg. The design team noted that, in the end, additional human trials should be made to ensure the safety and functional properties of the product with the selected dose of CSGH.

The first recipe of the food snack (Appendix A) and its production process (Appendix A) are presented in the Appendix A.

### 3.9. Creating the Prototype and Its Testing

During this step, a second prototype of the food product should be prepared according to the information gathered in Step 8. The prototype should then be subjected to sensory analyses as well as laboratory analyses to test its functional properties. The analysis of the technological feasibility and economic viability of the production, along with market research, should also be performed. Such analyses should result in obtaining data about the strengths and weaknesses of the current prototype, which allows for the modification and further improvement of the prototype in the next steps.

Case study: The second prototype was created and subjected to consumer sensory analysis, performed on 20 consumers. The prototype did not receive very high scores, with individual parameters marked within the range of 5 to 6 on a nine-point hedonic scale, with the exception of odour, which was only 4.3, below the acceptable threshold.

The cost of the raw materials to produce a date bar according to the proposed recipe is approximately 0.27 EUR. In FRAP analysis, it was shown that the antioxidant power of the prototype was improved by modification of the recipe for the preliminary prototype, reaching 13.90 ± 0.42 µM trolox/mg.

During consumer analysis at this stage of the design, the consumers were also asked to evaluate the stimulus intensity (taste, odour, colour, appearance, texture, size) of the product on a nine-point scale and to mark the stimulus intensity that the ideal product should have according to them. This allowed the design team to better understand which direction the changes in the prototype should go.

During this step, the design team performed market research on the available competition, considering 46 different date bars available on the market. The typical date bar had a weight of 35 to 50 g, and a price of 0.52 to 1.25 EUR for the final consumer (excluding VAT). The price differed mostly with regard to the types of ingredients used, as well as the market share of the product, with products offered through retail chains being cheaper than bars offered through less conventional distribution chains (e.g., internet shops). The main advantages and disadvantages of the newly developed product compared to the competition were:

Based on the results, the following weaknesses were determined: (i) the addition of carp meat changed the colour of the bar to light brown, meanwhile, the preferred colour marked by consumers was dark brown; (ii) fragmentation of the ingredients was too high, with consumers indicating they would prefer to see some ingredients such as nuts and seeds; (iii) the fishy flavour and odour was detectable and caused unfavourable results for sensory scores; (iv) the consumers indicated that the ideal date bar should be sweeter; (v) the cost of raw materials was high, and if possible, should be reduced.

The identified strengths included: (i) a source of complete protein; (ii) a source of omega-3 fatty acids; (iii) the higher weight of the individual date bar; (iv) the product meeting the current trends among consumers (gluten and lactose-free, without preservatives, etc.); (v) very high antioxidant power, confirmed in laboratory testing; (vi) the nutritional value of the product allows it to be fully sustainable and a separate meal which can replace many food supplements; (vii) a relatively simple production process without the need for excessive investment costs.

### 3.10. Modification of the Prototype and Re-Testing

Knowing the weaknesses and strengths of the product, the design team should try to analyse them and eliminate as many weaknesses as possible, while maintaining the strengths. This can be done by modifying the recipe or the production process of the product. After modification, the prototype should be tested once more. The tests may be the same as in Step 9 or include any additional analysis that might arise from the weaknesses and strengths. Step 10 should be repeated until the design team decides that no additional improvements can be added and a satisfactory level of quality has been obtained.

Case study: After the analysis of the strengths and weaknesses, the design team decided to modify both the recipe and production process of the date bars with carp meat. To reduce the impact of the carp on the sensory qualities of the product (colour, odour, sweetness), prototypes with different levels of carp meat within the range of 20 to 40% were produced and subjected to sensory and antioxidant power analysis. Based on the results, the amount of carp was reduced from 40 to 29%, while the levels of dates, buckwheat flakes, pecan nuts, and water were increased (Appendix A, Appendix A). Aside from the sensory scores, this modification also resulted in a 9.8% decrease in the cost of raw materials, since carp meat was the most expensive ingredient aside from pecan nuts and CSGH. The design team also modified the production process of the date bars (Appendix A—Appendix A), and switched the process of bowl cutting to mixing, which resulted in visibly larger particles of pecan nuts as well as sunflower seeds and improved texture.

The final version of the prototype was subjected to sensory analysis by the sensory panel. The snack received a sensory score of 4.09 on a five-point scale and the profiling analysis (Figure 2) showed that the fish odour was replaced by dried fruit/nuts and cinnamon aromas, while the fish aftertaste was eradicated using dried fruit/nuts, causing the taste to become sweet.

The panellists also noted that the structure of the date bar was no longer homogenous with clearly visible particles and a darker colour. At the same time, the antioxidant power showed no statistical changes in FRAP value (12.79 ± 0.24 µM trolox/mg) and a high DPPH radical inhibition rate (87.57 ± 0.37%).

### 3.11. Determining Nutritional Value, Shelf-Life and Storage Conditions

When the final formulation of the food product is obtained, it is necessary to analyse its nutritional value and establish the shelf life and storage conditions. Knowledge about the nutritional value of functional foods is essential, since those products are usually marketed for their health-promoting properties. The determination of the shelf life and storage conditions is broadly described in the literature [44,45,46]. In the case of food products with long estimated shelf lives, protocols for accelerated shelf-life testing can be introduced [47]. Depending on the food product, different types of analyses should be performed to establish the shelf life.

First of all, the design team should consider the necessary analysis required by local legislation. Moreover, the main causes of food spoilage include microbiological spoilage, oxidation, and autolysis [48]. Therefore, the design team should also consider performing various additional analyses to assess the microbiological quality (e.g., total viable counts, yeasts and mould, etc.), oxidation rate (e.g., TBARS, peroxide value, etc.), and quality changes (e.g., sensory evaluation, texture and colour, pH, etc.) during product storage. The storage conditions chosen for testing should be carefully considered. These usually include parameters such as temperature, humidity, light exposure, etc. Generally, for the majority of food products, the lower the storage temperature, the higher the shelf life. On the other hand, lowering the storage temperature causes problems related to storage and logistics, increasing the costs of those processes as well as the risk of mishandling by final consumers, resulting in impaired safety of the product.

Case study: Since the date bar with carp meat and CSGH is a product designed for physically active people, it is important to determine its nutritional value, including its mineral composition as well as the occurrence of certain vitamins, and to compare them with dietary recommendations. In (Table 1), the results of nutritional value analyses are shown.

According to recommendations, athletes practicing endurance and strength–endurance disciplines should have an intake of 30 to 60g of carbohydrates per hour of training [49]; therefore, a 70-g serving of the newly designed snack provides an appropriate amount of carbohydrates. The energy share from individual ingredients is 17.5% from proteins, 22.3% from fat, and 60.3% from carbohydrates. The amino acid composition influences the response of muscle protein balance following resistance exercise. Essential amino acids are necessary to acutely stimulate muscle protein and net muscle protein synthesis [50]. When compared to reference essential amino acid profiles [51], the only limiting amino acids in the snacks’ protein profile are methionine and cysteine; however, it should be noted that no data on the levels of tryptophan were collected.

Group B vitamins are crucial in maintaining health among physically active people, supporting energy production processes. Among athletes characterised by a low or even marginal nutritional status of B-vitamins, a decrease in the ability to perform highly vigorous exercise has been observed [52]. Athletes performing endurance sport disciplines are at an increased risk of B-vitamin deficiencies [53]. The innovative snack was also a good source of vitamins B1 and B12, with one bar daily covering 38.5 and 52.5%, respectively, of the recommended daily intake (RDI) for adult males. On the other hand, the RDI for vitamin B2 covered by one bar was only 3.8% [54]. The snack also provides a significant amount of Mn, Fe, Mg, K, and Zn, totalling 26.1, 15.2, 15.7, 7.8, and 8.9% of RDI from one 70 g date bar. In contrast, it is not a good source of Ca (0.3% of RDI) or Na (0.5% of RDI) [54].

The shelf life of a food product can be defined as the amount of time during which a product still exhibits the same level of quality during storage in defined conditions. It also requires finding the main discriminating parameters, with acceptability limits established for each parameter [55]. The main quality parameters with acceptability limits used to establish the shelf life in this study were: (i) microbiological contamination, with the upper limit for TVC of 6 log CFU/g, as established by Gilbert et al. [56]; (ii) stable antioxidant activity, without a significant decrease in both DPPH and FRAP; and (iii) consumers’ scores of the product above the established acceptability threshold of 5.

During shelf-life evaluation, the date bars with carp meat and CSGH stored at room temperature became covered with mould after 10 to 14 days of storage. Due to this, room temperature storage was discarded by the design team. On the other hand, the microbiological quality of snacks stored in chilled and frozen conditions remained stable throughout the whole storage period (six weeks for chilled and six months for frozen storage) (Figure 3).

The initial TVC and YM of the snack was 4.43 and 3.08 log10 CFU/g, respectively, and the storage did not cause statistically significant increases, neither in TVC nor YM; therefore, the snacks were greatly below the established acceptability limit. The initial water activity (aw) of the samples was 0.834 ± 0.000 and did not change significantly during storage. This means that the designed snack is an intermediate moisture food (IMF), for which aw is generally from 0.6 to 0.85 in room temperature. IMFs containing protein hydrolysates are prone to the occurrence of moisture-induced Maillard browning during storage if reducing sugars are present. This leads to the loss of free amino groups in IMFs [57]. The loss rate can be increased significantly during storage due to increased molecular mobility from using protein hydrolysates. This quality loss may eventually lead to the loss of the claimed bioactivity of protein hydrolysates in the products, which is why, during the design process of new functional products, with the addition of protein hydrolysates, stability evaluation of their biological properties and quality during storage is necessary.

The changes in antioxidant activity of the developed date bars are shown in (Table 2).

As previously mentioned, the newly designed snack had high radical scavenging properties (DPPH) and FRAP power. The obtained values of antioxidant properties of the designed bars are high compared to data presented in the literature. Kaur et al. [58] determined the antioxidant properties of gluten-free cereal bars at the level of 48 to 78%. Parn et al. [59] produced date bars from a variety of these fruits, the antioxidant activity of which was also relatively low, ranging from 30 to 53% (tested via the DPPH radical scavenging method). Additionally, Reis and Abu-Ghannam [60] indicated the moderate antioxidant activity of bars produced with brewer’s spent grain. The value of the FRAP parameter for these products ranged from 50 to 350 µg ascorbic acid/g of dry weight, while the ability to scavenge DPPH free radicals by the antioxidants contained in the bars was at the level of 20 to 65%.

During storage, the DPPH radical scavenging properties fluctuated, which was most probably related to differences between individual samples. However, the DPPH decreased significantly in both storage conditions during the final week of storage in chilled conditions and after 17 weeks of frozen storage. FRAP decreased significantly and progressively during the last weeks of storage in chilled conditions, but remained stable during frozen storage, with fluctuations observed at week eight of storage. The snack ingredients and CSGH can react with one another, leading to unpredictable changes in antioxidant activity, with implemented processing procedures further influencing the shape of those changes. This is mainly due to redox reactions between antioxidants or antioxidants and fat oxidation products [58].

The TBARS analysis showed slow but significant progress in the lipid oxidation process, increasing from the initial value of 10.73 mg TBARS/kg to 15.19 and 17.36 mg TBARS/kg, respectively, at the end of the chilled and frozen storage periods (Table 2). However, there were no observed significant differences in TBARS values between chilled and frozen storage. In addition, it has been observed that the degree of lipid oxidation in the snack increases as its antioxidant properties decrease during storage.

Consumer sensory analysis performed on 50 consumers after six weeks of chilled storage showed that the sensory scores for all parameters of date bars with carp meat and CSGH remained above the acceptability threshold for the whole storage period, with the overall quality score of 6.2 on a nine-point scale.

Based on the acquired results, the design team decided that the newly developed product has to be stored at a controlled temperature—either chilled or frozen—to maintain its safety and properties. Microbiological stability of the snack for at least six weeks at 4 °C was considered a success considering the snack contained high quantities of fish meat. Due to technological, economical, and logistical issues related to frozen storage, the design team decided that the 4 °C storage temperature is the most suitable option; however, frozen storage remains a possibility for the future, should a longer shelf life be required.

However, due to a significant reduction in antioxidant power during chilled storage from week four onwards, the shelf life of the final product has been established at four weeks of storage at 4 °C.

### 3.12. Determining the Final Product Characteristics and Preparing the Necessary Documentation Including the GMP/GHP and HACCP Guidelines

The final step of the design protocol was to prepare documentation related to the final characteristics of the product. This includes updated recipes, production process, storage conditions, etc. The additional report should include the results of all the analyses, comprising the identified strengths and weaknesses of the product as well as suggestions for future improvements. The documentation should also provide information on the framework for food safety management systems, i.e., HACCP and HARPC, and GMP/GHP rules related to product production. These may include hazard analysis, suggested preventive measures, the identification of possible critical control points (CCPs), instructions related to the production process, identification of possible allergens with guidelines for their handling, etc. The design team should also try to perform a classification of the designed product, taking into account the legislation requirements for the products according to the selected food product class. Those can include periodical microbiology analyses, analyses for residues of some contaminants or others and should be included into the framework of the prepared food safety management system. This allows for easier implementation of the newly developed product into full-scale production.

Case study: As the last step, the design team prepared a documentation package, which included recipes with production technology, a product description sheet, and a hazard analysis along with CCP identification. Five CCPs were identified: CPP1—receiving goods, CPP2—raw material storage, CPP3—thermal treatment of fish meat, CCP4—final product storage, CPP5—expedition. The product has been classified as a ready-to-eat product, recommended for people who perform regular physical activity. The design team has performed a review of the existing legislation within the European Union, and decided to include additional requirements into the documentation package. The date bars with carp meat and CSGH can be classified according to EU Regulation 2073/2005 as a ready-to-eat food able to support the growth of *Listeria monocytogenes* other than those intended for infants and for special medical purposes, which obliges the manufacturer to perform periodical testing of the product for this bacterium. No other contaminants, as described in EU Regulation 1881/2006, were found to be necessary for periodical testing, since the analysis for those is the obligation of the suppliers of raw materials, such as nuts, that will be used in the final product. Based on the EU Regulation 1169/2011, the label of the product should contain information about three allergens: fish (from carp meat and possibly CSGH), gluten (from buckwheat), and nuts (from pecan nuts). The analysis of the product, performed in the previous steps, also provides complete information regarding the nutritional parameters, which are also required to be placed on the label of the product. The documentation also included information about the necessary safety studies mentioned in Step 3.8, which also needs to be conducted if the product is to be introduced into the European Union. The documentation packages were included into the overall report documenting the whole design process and were presented to the research facility along with the final version of the newly developed date bar with carp meat and CSGH.

### 3.13. Limitations of the Study

The newly developed protocol as well as the described case study has its limitations. The aim of the protocol was to allow simplification and ease of adopting the design thinking approach to design new functional food products, which would allow the use of this approach by designers who are inexperienced or unfamiliar with design thinking. This resulted in the formation of a step-by-step protocol, which can be interpreted as linear, while the design thinking is, at its core, a non-linear approach to design. Therefore, the protocol showed in Figure 1 could include more branching and backwards pathways; however, in the opinion of the authors, this would greatly impair its readability and make it more difficult to use and understand, which, in turn, would impair the aim of the protocol. Thus, the design team should always bear in mind that the protocol can finish in reaching a dead end, and the design team should not be afraid of going backwards, even if this means coming back from the final testing phase to the empathize step.

The second limitation is related to the described case study. Although design thinking should be purely human-centred, the presented case study started with certain requirements towards the developed product, which limited the approach of fully unlimited designing (the product must contain carp meat, and must contain CSGH). In an ideal situation, the design team should not have any preliminary requirements, and all of the product characteristics should be based solely on the identified needs of the consumer target group. However, it should be noted that, in most cases, such an approach is not possible because the design of new functional products is usually ordered by some business entity that is already present on the market, and this entity often does have certain initial requirements to be considered. These may include the use of specific raw materials, technology, machines, etc. due to limiting the costs of investment which has to be made during the implementation of the newly developed product and usually depends on the financial capabilities of the entity. In this case study, the customer was actually an industrial carp processor; however, the protocol ensured that the main focus of the design was firstly on the potential user and not on the product itself.

## 4. Conclusions

In this study, a newly developed 12-step protocol has been described for designing novel functional food products with the addition of food industry by-products. The protocol is based on the design thinking principles, and by splitting them into 12 clear steps, they should be easy to follow and implement into almost any food production sector. The protocol has been successfully used to design and develop a new functional food product: date bars with carp meat and CSGH, designed as a peri-workout snack for physically active people. The product addresses the identified needs of the target consumer group and has very high nutritional value as well as antioxidant activity, which remains stable during storage at reduced temperatures. Moreover, the product is characterised by good sensory qualities and can be easily implemented into full-scale production. The newly designed protocol can be used to successfully develop user-focused functional food products.

## Figures and Tables

**Figure 1 foods-10-00694-f001:**
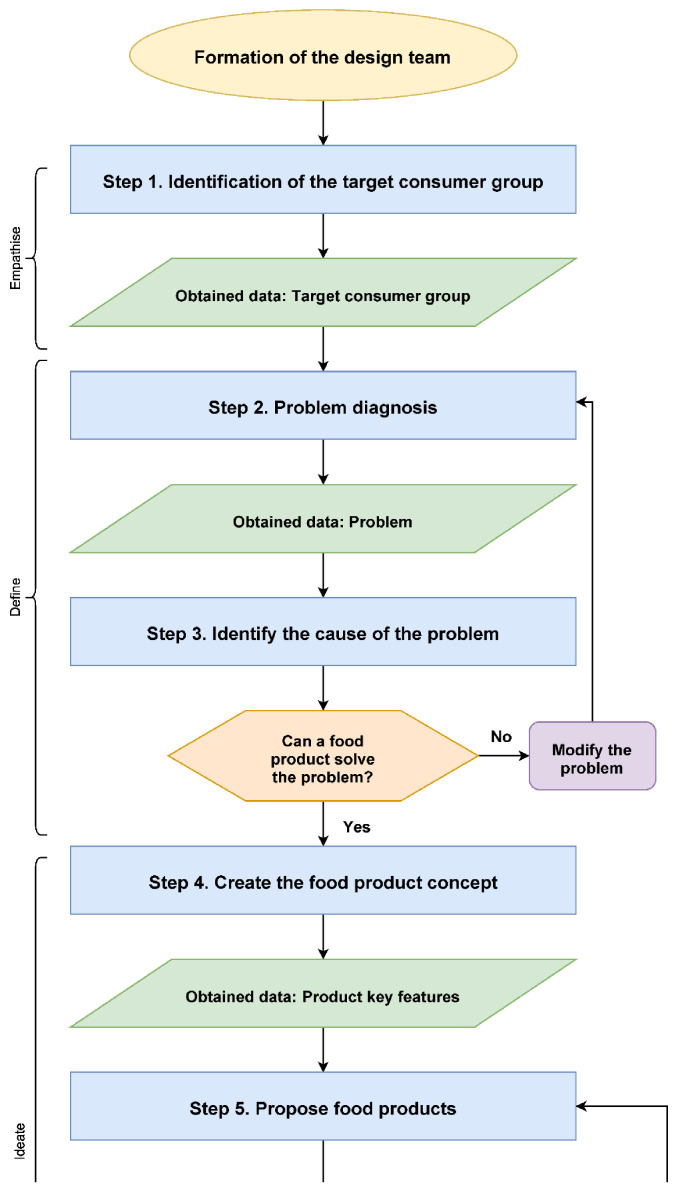
Protocol for designing new functional food with addition of food industry by-products using the Design Thinking methodology.

**Figure 2 foods-10-00694-f002:**
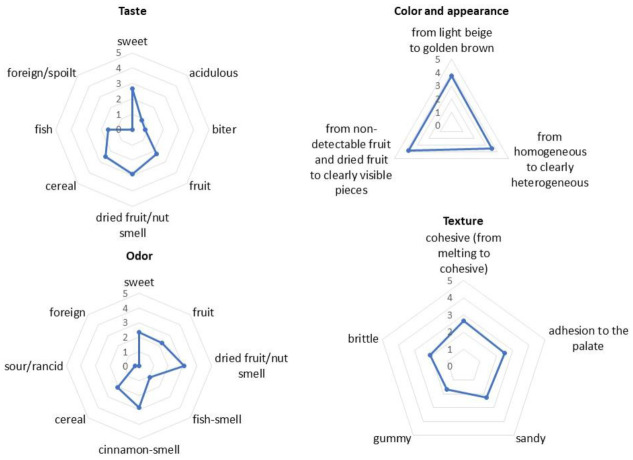
Sensory evaluation of innovative snacks via profiling analysis.

**Figure 3 foods-10-00694-f003:**
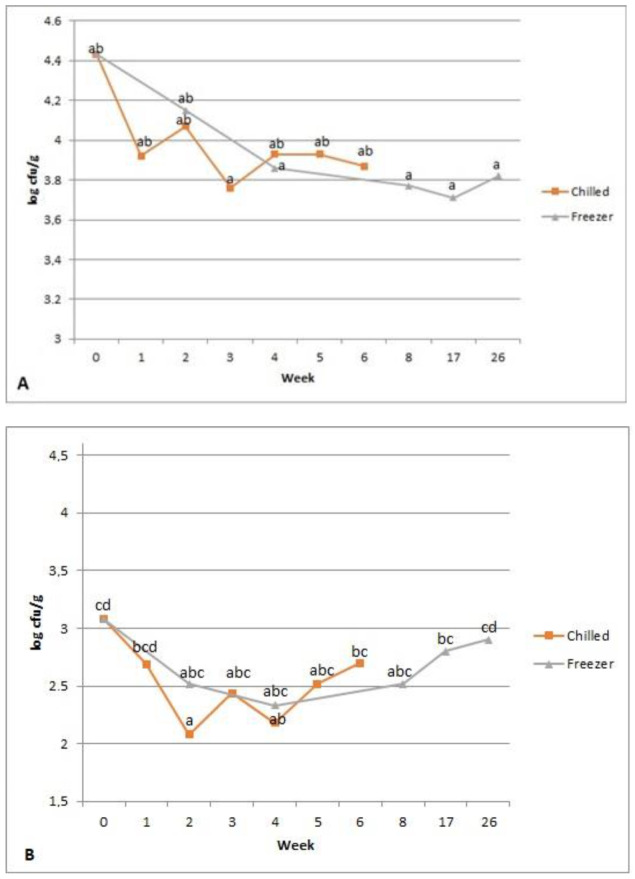
Microbiological quality of innovative snacks with CSGH stored in different conditions. (**A**) Total viable aerobic (TVC) count, (**B**) yeast and mold (YM). Results presented as mean; ^a,b,c,d^—results marked with varied lettering are significantly different.

**Table 1 foods-10-00694-t001:** Nutritional value of the date bar with addition of fish industry by-products designed according to new protocol.

**Nutrient**	
Dry weight [%]	70.69 ± 0.13
Protein [%]	12.39 ± 0.30
Fat [%]	6.98 ± 0.13
Ash [%]	1.84 ± 0.04
Carbohydrate [%]	42.69 ± 0.22
Dietary fibre [%]	6.78 ± 0.90
Cholesterol and plant sterols (mg/kg)	580.7 ± 0.9
**Amino Acids [%] of Protein**	
Aspartic acid	13.24 ± 0.05
Serine	4.63 ± 0.26
Glutamic acid	20.01 ± 0.22
Glycine	5.83 ± 0.34
Histidine	3.47 ± 0.22
Arginine	7.13 ± 0.11
Threonine	3.98 ± 0.23
Alanine	3.44 ± 0.20
Proline + hydroxyproline	1.89 ± 0.08
Cysteine	0.29 ± 0.07
Tyrosin	2.18 ± 0.07
Valine	5.81 ± 0.07
Methionine	0.86 ± 0.03
Lysine	10.14 ± 0.13
Isoleucine	5.11 ± 0.02
Leucine	7.91 ± 0.08
Phenylalanine	4.08 ± 0.12
**Vitamin [µg/kg]**	
Thiamine	6613 ± 395
Riboflavin	672 ± 37
Cyanocobalamin	18.0 ± 0.2
**Minerals [mg/kg]**	
K	5255 ± 84
Na	158.5 ± 1.2
Ca	43.4 ± 4.6
Mg	900.0 ± 16.6
Mn	8.6 ± 0.5
Fe	17.5 ± 0.5
Zn	14.0 ± 0.0

Results presented as mean ± SEM; dry weight, protein, fat, ash, carbohydrates, dietary fibre are expressed in % *w*/*w*; cholesterol and minerals are expressed in mg/kg of product; vitamins are expressed in µg/kg of product; amino acids are expressed as % of protein.

**Table 2 foods-10-00694-t002:** Results of the shelf-life determination of the date bar with addition of fish industry by-products designed according to new protocol.

Week	DPPH [%]		FRAP [µmol Trolox/mg]		TBARS [mg Malonaldehyde/kg]	
	C	F	C	F	C	F
0	87.57 ^d,e^ ± 0.37	87.57 ^d,e^ ± 0.37	12.79 ^b,c^ ± 0.24	12.79 ^b,c^ ± 0.24	10.73 ^a^ ± 0.43	10.73 ^a^ ± 0.43
1	88.03 ^e^ ± 0.62	x	13.61 ^b,c^ ± 0.18	x	13.03 ^a,b^ ± 0.23	x
2	77.46 ^a,b^ ± 0.81	78.58 ^a,b,c^ ± 0.41	12.79 ^b,c^ ± 0.31	13.67 ^b,c^ ± 0.17	11.85 ^a,b^ ± 0.72	11.45 ^a^ ± 0.42
3	82.34 ^a,b,c,d,e^ ± 0.24	x	13.09 ^b,c^ ± 0.12	x	14.32 ^b,c^ ± 0.45	
4	80.77 ^a,b,c^ ± 0.44	81.06 ^a,b,c,d^ ± 0.33	12.70 ^b^ ± 0.21	13.28 ^b,c^ ± 0.33	13.08 ^a,b^ ± 0.40	13.92 ^a,b^ ± 0.47
5	83.40 ^b,c,d,e^ ± 0.23	x	9.50 ^a^ ± 0.32	x	12.46 ^a,b^ ± 0.27	x
6	69.28 ^a^ ± 6.01	x	5.85 ^a^ ± 0.23	x	15.19 ^b,c^ ± 0.38	x
8	x	85.32 ^c,d,e^ ± 0.29	x	10.54 ^a^ ± 0.37	x	16.03 ^b,c^ ± 0.45
17	x	74.07 ^a,b^ ± 3.42	x	13.62 ^b,c^ ± 0.22	x	16.08 ^c^ ± 1.01
26	x	74.96 ^a,b^ ± 3.24	x	13.98 ^c^ ± 0.20	x	17.36 ^c^ ± 0.52

Results presented as mean ± SEM; ^a,b,c,d,e^—results marked with varied lettering are significantly different; C—chilled, F—frozen, x-analysis not performed during this day.

## Data Availability

Not applicable.

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
