# Peer review of "Protocol for Designing New Functional Food with the Addition of Food Industry By-Products, Using Design Thinking Techniques—A Case Study of a Snack with Antioxidant Properties for Physically Active People"

_foods, 2021, doi:10.3390/foods10040694_

Round 1

Reviewer 1 Report

The paper foods-1117345  developed a protocol with twelve steps to design a novel functional food with the use of food industry by-products. The protocol is based on the Design Thinking principles, and it is easy to follow and implement. The protocol has been successfully used to design and develop a date bars with antioxidant properties containing carp skin gelatin hydrolysate.

The experimental work is well described and includes the determination of different food nutrients/food components by validated methods.

The discussion of the results is consistent, therefore I recommend minor modifications. The only recommendations are the following:

- Line 131: Please indicate the name of the first author of the reference 22

- Equation (2), please change 0,2 by 0.2

- Figure 3: please increase the size of the numbers of xx and yy axes and change the decimal separator from ‘,’ by ‘.’

- Table 2: check if the results of TBARS should not be in mg of malonaldehyde/kg

- In the conclusions section please indicate which are the consumer need when you mention “The product addresses two identified needs of the target consumer group”

Reviewer 2 Report

The manuscript reports on the development of a date bar that contains fish by-products. The approach is based on Design Thinking and is supported by sensory assessment, microbiological test and chemical analyses. The manuscript is very well written and clearly points out the advantages of the authors’ approach. Each step is briefly introduced and the implementation in the case study is reported. In my opinion the manuscript could almost be published in its present form. However, I have some comments, questions and suggestions, that can be interesting to include.

Chemical analyses: The authors have made huge effort to chemically characterize the product. I suggest to include detailed information about sample treatment like extraction protocols etc. in the supplement.

As it seems the product was mainly tailored for male consumers with a body weight of 70 kg (lines 472, 597). Could the authors include a short comment on how this product would be suitable also for female persons and people carrying more or less weight?

Would the product or its ingredients, especially the skin hydrolysate, have to be considered as Novel food?

The aim of the process was to develop a product that solves certain problems of physically active people. The product has now been developed and I wonder if the consumers would really notice a positive effect. A short outlook on how such a study would have to be designed or is planned would be interesting. Would it be possible at all to gain scientific evidence related to whether the date bar meets the expectations?

At the end of the storage period the sensory properties were just above “acceptable”. Would this score be high enough for consumers to actually buy this product (more than once)? Do you have any information on the sensory scores of similar products that are on the market?

Regarding the DPPH, FRAP and TBARS results: Are the achieved values high / low / comparable to other snacks like cereal or protein bars? A comment on the classification would be interesting.

Reviewer 3 Report

In the study, a newly developed 12-step protocol is described for designing novel functional food products with the addition of food industry by-products. The protocol is based on the Design Thinking principles, and by splitting them into 12 steps.Stepd described should be easy to follow and implement into almost any food production sector.

The paper is well described and almost all steps of production have been included.

I am not sure and I ask to better describe if a functional food need to be approved before selling (as a novel food or supplement), furthermore I think that it is necessary to include legislation requirement in the text. Authors report all the analisys done to asses quality of product and safaty but they don't report which are the requirment needed and regulation in force.

Author Response

Comment: I am not sure and I ask to better describe if a functional food need to be approved before selling (as a novel food or supplement), furthermore I think that it is necessary to include legislation requirement in the text. Authors report all the analisys done to asses quality of product and safaty but they don't report which are the requirment needed and regulation in force.

Response: In the case study, we have focused on general quality and safety analyses, mainly due to the fact that legislation regulations differ from country to country and vary in the USA, China, the EU, Japan, Australia or New Zealand etc. We also admit that we are unaware of the details regarding legislation in most of those countries. Meanwhile, the quality aspects of functional foods are basically constant in all parts of the world. The Reviewer, however, has made an excellent point and information about legislative issues should be included in the manuscript. 

As for approvement of the new product in the European Union, it will mostly depend on how we treat the carp skin gelatin hydrolysate (CSGH). If we consider it as a “food”, then, we will have to obey the Novel Food regulations (Regulation (EU) 2015/2283 and its future amendments). As is stated in this Regulation, Novel Food is defined as that which has not yet been consumed to a significant degree by humans in the EU before 15 May 1997 (the date of the first regulation on the matter of novel food came into force in the EU). Since CSGH has been developed by our team in 2018, then by definition, it is ‘novel’, because it could not have been consumed before 1997. This means that in order to actually place the product on the EU market, the company would have to first go through quite tiresome process of listing the newly developed product within the Novel Food Catalogue. According to our knowledge, there is no exception for protein hydrolysates in this matter, even though they are generally considered safe.

CSGH could also be considered not as a “food” but as a “food additive” – however, in this case it would also have to go through a rigorous process of authorisation, laid in EU Regulation 1331/2008, so that it could be listed in Regulation 1333/2008. This would be a more appropriate path if we considered using the CSGH alone for different products and not as part of the date bars with CSGH described in this case study.

Paradoxically, the date bar with CSGH could also be sold not as a food product – but as a food supplement. This way, the product would have to only be reported and submitted on a national level, and not go through the EFSA procedures, making it much easier to launch a product on to the market, since food supplements can be sold in retail chains alike regular food products. The only difference would be in labelling the product – since it would have to state that this is a food supplement, with the recommended daily dose and all the other appropriate information listed in 2002/46/EC with further amendments. The question remains whether such bypassing regulations would be ethical or not. This, nonetheless, I believe is matter for a different and much broader debate. An ideal example of differences between novel foods and food supplements, and legislation problems around those two, is hyaluronic acid – it is one of the most commonly sold food supplements in Europe with practically no established upper limit doses. It can be sold in powder form, pills, droplets or even products containing hyaluronic acid (but labelled as food supplements). Meanwhile, it is not possible to be used as part of actual food products because it is not listed as a food additive in regulation 1333/2008. Moreover, the Bioiberica company has developed a rooster comb extract which contains mostly hyaluronic acid (market name Mobilee), and this compound is actually listed on novel food lists. Thus, rooster comb extract can be used, hyaluronic acid can be consumed in any amounts in pills, but it cannot be added as pure hyaluronic acid to, for example, sausage or other food products. 

Therefore, all of the-above, may actually be possible paths not only for the product presented in this case, but for most functional foods. In my opinion and experience, the procedure is much easier when introducing products as food supplements than as novel foods.

We have added more information about the need for legislation issues into the protocol (Step 7) and its description. We have also provided data about this into the description of the Case Study, however, we did not elaborate much on the legislative problems of the EU since the manuscript is designed to be applicable to readers across the world, and the legislation, for example, in the USA is different. Meanwhile as we explained using the hyaluronic acid example, the legislation status of food products in the EU is very complicated, and it would probably be enough for writing a separate review article. Therefore, since the manuscript is already quite long and the case study is meant only as an example for showing how to operate, we decided to add only simple information about the necessity of applying for novel food to the European Commission and added a reference to a EFSA article in the EFSA Journal which. in detail, provides and explanation on the whole procedure.

Round 2

Reviewer 3 Report

I think that the article must be improved and that it is not possible to describe a particular case study without describing all the needed point to put it on market. It is necessary to classify it or at leat declare how they suppose to classify it.

It is essential to describe all the point to set up a new product for marketing. If a specific case study is reported it is essential to describe how the new product is classified and what are the legislative parameter to be considered to put it on market. I know that legislation in force it is not common on all the world,
but some parameter are necessary and also the classification of the new product is essential. A case study without classification it is not a scientific case study.

Author Response

I would like to thank you and the Reviewers for the time and effort invested in reading and correcting our manuscript in a thorough manner, and also for the comments which led to improvement in the manuscript quality. We have revised the manuscript in light of the suggestions and comments. We hope that the revision has improved the paper to a level of satisfaction. Red colour in the text indicates corrections or modifications to the manuscript. Detailed responses to specific comments/suggestions/queries are listed below:

Comment: I think that the article must be improved and that it is not possible to describe a particular case study without describing all the needed point to put it on market. It is necessary to classify it or at leat declare how they supposed to classify it.

It is essential to describe all the point to set up a new product for marketing. If a specific case study is reported it is essential to describe how the new product is classified and what are the legislative parameter to be considered to put it on market. I know that legislation in force it is not common on all the world, but some parameter are necessary and also the classification of the new product is essential. A case study without classification it is not a scientific case study.

Response: We have modified the information in Step 3.8 adding additional paragraph on the novel food requirements and added the information about the classification of the product in the last step (Step 12). We have added information about the revision of local legislation along with the classification of the product in general and in accordance to those legal rules. We have mentioned additional microbiological parameter that has to be periodically tested, included information about the labeling requirements.
